# Digital Examination of Vegetation Changes in River Floodplain Wetlands Based on Remote Sensing Images: A Case Study Based on the Downstream Section of Hailar River

Xi Dong [1,2] and Zhibo Chen [1,2,*]

1   School of Information Science & Technology, Beijing Forestry University, Beijing 100083, China; dx_bjfu@126.com

2   Engineering Research Center for Forest-Oriented Intelligent Information Processing, National Forestry and Grassland Administration, Beijing 100083, China

\*   Correspondence: zhibo@bjfu.edu.cn

**Abstract:** The Hailar River is an important river in the Inner Mongolia Autonomous Region, China. It plays an extremely important role in maintaining the ecological balance of the region. However, in recent decades, the Hailar River and its surrounding areas have been developed at a high rate and its wetland resources have faced various threats. In this study, vegetation changes in the Hailar River wetlands were analyzed using remote sensing data from the Landsat TM (1987, 2001, and 2010) and Landsat OLI-TIRS (2019) satellites. A vegetation change model was developed using Matlab software to assess vegetation changes in the area. There were significant changes in the wetland vegetation of the lower Hailar River study site between 1987 and 2019. There was an increase in open sand habitat with a sparse vegetation area of 1.08 km$^2$, a decrease in grassland area of 13.17 km$^2$, and an increase in the forest area of 15.91 km$^2$. The spatial distribution of the normalized difference vegetation index (NDVI) varied across the study site and was high overall. The vegetation types varied with distance from the river. There are two possible explanations for positive and negative vegetation change trends. In areas where the water supply is sufficient and relatively stable, the cover of forest vegetation was gradually increasing and the herbaceous plant community is gradually evolving into a scrub woodland plant community. In areas where the water supply is lacking, there are changes in the sense of a decrease of forest vegetation and an increase of open sand habitat with sparse vegetation. Therefore, this study suggests that the existing wetlands should be protected, used wisely, and developed rationally to provide sustainable resources for the next generation.

**Keywords:** Hailar River; remote sensing; river floodplain wetlands; vegetation index; vegetation changes

## 1. Introduction

River floodplain wetlands are valuable natural resources for human beings. They have the functions of regulating regional ecological conditions and environments and provide irreplaceable direct and indirect services to humans [1–3]. In recent years, rivers and river floodplain wetlands around the world have undergone change [4–7]. These changes greatly affect terrestrial and aquatic environments, such as by altering wildlife ecosystems and threatening regional sustainable development [8,9]. Therefore, the study of vegetation changes in river-floodplain wetlands is key to their conservation and the management and planning of wetland resources.

There are wetlands on both sides of the main section of the Hailar River, which are key to reducing ecological risks and building ecological patterns in the river basin. They are critical ecological corridors linking the elements of the river basin's ecosystem. These ecological corridors are extremely important areas linking the water-containing ecological functions of the Greater Khingan Range and are also ecological security maintenance areas

for biodiversity protection, climate regulation, and food and water supplies [9]. They are of great importance to regional wind control and sand fixation, vegetation restoration, and the construction of ecological communities comprising mountains, water, forests, lakes, and grasslands, which cover the whole watershed and its elements.

In recent years, remote sensing technology has developed rapidly, allowing researchers to gain its advantages of cost- and time-savings for the monitoring of wetland vegetation. This has made it possible to monitor long-term vegetation changes in river-floodplain wetlands [10,11]. The use of remote sensing (RS) and geographic information system (GIS) techniques to study vegetation changes along river floodplain wetlands is an ongoing focus of research on regional and global scales. This includes monitoring of land cover classification, vegetation changes, water use, and species diversity [11].

More than 100 vegetation indices are currently in use, especially the normalized difference vegetation index (NDVI), which is the most commonly applied indicator in environmental and climate change studies [12]. The NDVI is normalized by the relationship between the near-infrared (NIR) reflectance of a given remotely sensed image and the red band (RED) [13]. Many researchers have used NDVI to assess crop cover [13], vegetation [13], and agricultural drought [14]. Vegetation indices are simple measurement parameters used for environmental resource management and provide information that is useful for assessing the Earth's plant cover and crop growth at the pixel level based on satellite images [15]. These methods provide different ideas for the study of wetland vegetation around rivers. However, few studies have been conducted on river-floodplain wetlands in the lower section of the Hailar River [16,17]. This study focuses on vegetation changes at the pixel level in river-floodplain wetlands in this region.

This study takes the downstream section of the Hailar River in the Inner Mongolia Autonomous Region, China, as the research object. Remote sensing images are used to explore vegetation changes in the river floodplain wetlands on both sides of the river. This study hopes to achieve the following main objectives: (1) To quantitatively assess vegetation changes and the spatial distribution of river-floodplain wetlands in the lower section of the Hailar River over the past 30 years; (2) to explore vegetation changes in the river floodplain wetlands, and; (3) to identify areas with significant vegetation changes in order to inform decision-makers and, thus, improve sustainable wetland planning policies.

## 2. Materials and Methods

### 2.1. Study Site

The Hailar River is located in Hulunbeier City, Inner Mongolia Autonomous Region, China (Figure 1). It originates at the western foot of the Greater Khingan Range in the territory of Urqihan Town, Yakeshi City, and is the upstream section of the Erguna River. It flows in an east-west direction (Figure 1) at a location of approximately 49.12° N and 119.39° E. The wetlands and swamps of the river are widespread, so that water can readily diffuse and seep during high water levels, resulting in a lower runoff volume downstream than upstream. The tributaries on the west bank are densely packed and the river network is dendritic in structure. The banks of the main streams and tributaries are covered with pristine forests and secondary forests with well-developed vegetation and strong water-supporting effects and form the main flow-producing area of the Hailar River.

The vegetation cover on both sides of the main stream of the Hailar River is relatively high and plant growth is vigorous. The vegetation type is mainly scrub, mixed scrub, and natural pasture, with relatively few trees. The scrub vegetation is mainly marsh willow (*Salix rosmarinifolia* L. var *brachypoda* (Traktv. Et Mey) Y. L. Chou), yellow willow (*Salix gordejevii* Y. L. Chang et Skv.), and brome (*Caragana sinica* (Buc'hoz) Rehder). The forests include coniferous forests (*Larix gmelinii* (Rupr.) Kuzen, *Pinus sylvestris* var. *mongolica* Litv.) and mixed coniferous and broad-leaf forests (*Larix gmelinii* (Rupr.) Kuzen, *Betula platyphylla* Suk., *Picea asperata* Mast.).

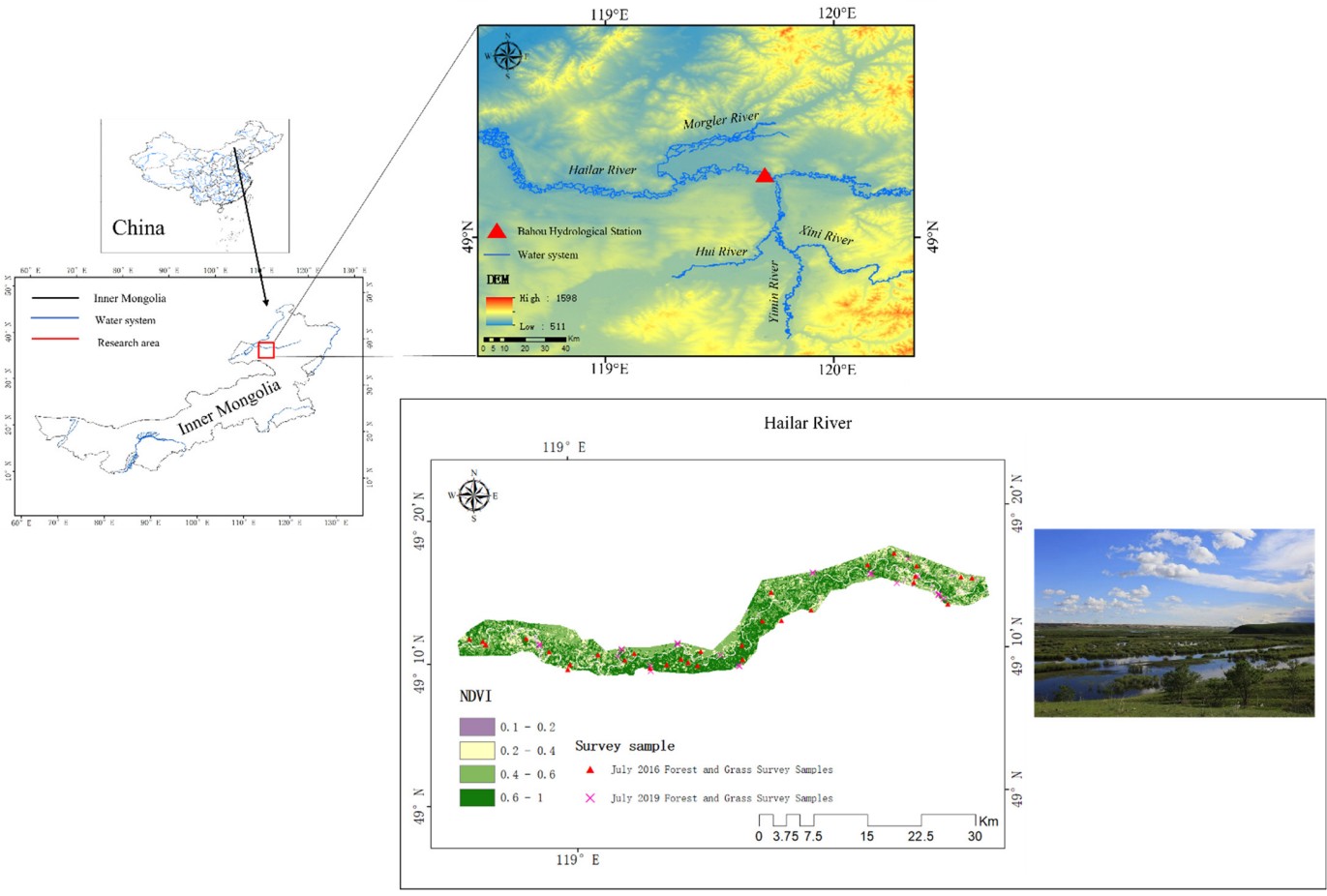

**Figure 1.** Location of the study site.

### 2.1.1. Data Sources

Current remote sensing satellite data provide diverse data options for the extraction of forest vegetation information. Considering the large area of the study site, Landsat 5 TM/8 OLI-TIRS 1T level images for this study were acquired from NASA (Table 1). The images cover the whole study site with the rank number 123,026 and were acquired in the summer growing period (July–August) of 1987–2019. Only images with <2% cloud were used. In this study, image data were acquired on 24 July 1987, 15 August 2001, 23 July 2010, and 16 July 2019, based on the key phenological periods of forest and grassland vegetation. Landsat provides medium- and high-resolution earth observations, which greatly reduces the data cost of large-scale remote sensing monitoring of forests and improves its timeliness. Data on the dominant tree species were obtained from the Hailar Regional Forestry Bureau.

**Table 1.** Landsat5 TM/8 OLI data situation.

| Satellite | Band | Nominal Spectral Location | Satellite | Band | Nominal Spectral Location |
|---|---|---|---|---|---|
| Landsat5 TM | 1 | Blue | Landsat8 OLI | 1 | Costal/Aerosol |
|  | 2 | Green |  | 2 | Blue |
|  | 3 | Red |  | 3 | Green |
|  | 4 | Near IR |  | 4 | Red |
|  | 5 | SWIR-1 |  | 5 | Near IR |
|  | 6 | LWIR |  | 6 | SWIR-1 |
|  | 7 | SWIR |  | 7 | SWIR-2 |
|  |  |  |  | 8 | Panchromatic |
|  |  |  |  | 9 | Costal/Aerosol |

2.1.2. Primary Data Processing

Radiance was calculated using the Radiometric Calibration tool (Radiometric Calibration) in ENVI 5.3, using an MTL.txt metadata file as input data. The output data was set to BIL format by line band. The output data type was set to floating-point data, the scale factor was automatically set to 0.1 and, to obtain the atmospherically corrected reflectance product, the radiation file was entered into the One-Angle Atmospheric Analysis of Hyperline-of-Sight (FLAASH) atmospheric calibration module. The Atmospheric Correction Module (ACM) of FLAASH in ENVI 5.3 was used, which automatically enters the appropriate correction parameters from the radiometric images. Other parameters (e.g., date, time-of-flight, and ground elevation) must be entered manually. We set the atmospheric model to Temperate and the Aerosol model to Rural, while the Aerosol retrieval was automatically set to 2-band K-T (Kaufman-Tanley method). The input parameters varied from scene to scene. The values of all reflectance products ranged between 0 and 1, and all of the obtained reflectance products were geometrically refined. The selected 10 alignment points were evenly distributed within the image experiment area and the errors were all less than 1. The normalized vegetation index was then calculated by the waveform calculation tool.

*2.2. Methods*

2.2.1. NDVI Vegetation Index

The NDVI was selected from the vegetation extraction model of remote sensing images. The NDVI is sensitive to the chlorophyll content of leaves and can be obtained using Equation (1) [18].

$$NDVI = (NIR - RED)/(NIR + RED) \tag{1}$$

NIR is the reflection value of the near-infrared band, RED is the reflection value of the red light band. Many researchers have used the NDVI for vegetation change monitoring (e.g., [19,20]). Through their studies, it was found that changes in NDVI can reflect vegetation changes.

The NDVI values were transformed into the range of ($-1$ to $+1$) so that the best classification features could be obtained by the best differentiation threshold. Tables 2 and 3 show the most common NDVI differentiation thresholds and corresponding categories according to the Remote Sensing Phenology Foundation [21] and [22]. According to the findings of [11,23,24], when extracting vegetation changes for a wetland study site, a uniform NDVI extraction threshold should be used. It has been shown that different NDVI vegetation extraction thresholds can be used for all classification extracted images.

**Table 2.** Base classification of normalized difference vegetation index (NDVI) values according to remotely sensed phenology [22].

| NDVI Range | Class |
|:---:|:---:|
| $\leq$0.1 | Low NDVI |
| 0.2–0.5 | Mid NDVI |
| 0.6–0.9 | High NDVI |

**Table 3.** NDVI values according to [22].

| NDVI Range | Class |
|:---:|:---:|
| 0–0.1 | Barren areas of rocks, sand or snow |
| 0.2–0.3 | Shrubs and grasslands |
| 0.6–0.8 | Temperate woodland and tropical rainforest |
| Close to 0 | Open sand habitat with sparse vegetation |
| Negative value | Water |

### 2.2.2. Vegetation Classification

In this study, the NDVI was used to show the distribution status of wetland vegetation. The NDVI values were extracted from four time series of Landsat images and classified using an iso-clustering unsupervised classification technique. The results were then used to compare the changes in wetland categories across the four time points of remote sensing images. The NDVI was reclassified into five categories: dense forest, sparse forest, grassland, open sand habitat with sparse vegetation, and no vegetation.

Vector data from a survey of the dominant tree species in forest land in 2016 were obtained from the Hailar District Forestry Bureau and combined with NDVI data extracted from the 2016 Landsat summer vegetation growth bloom period. By calculating the NDVI of the dominant tree species in the area of interest, NDVI classes were calculated as dominant tree species NDVI = 0.4–1, grassland= 0.2–0.4, and open sand habitat with sparse vegetation = 0.1–0.2, as shown in Figure 2. Combined with the NDVI classification thresholds provided by [22], this study classified NDVI into three categories as shown in Table 4: open sand habitat with sparse vegetation, grassland, and forest. The dominant tree species within sparse and dense forests in the study site had similar compositions and were dominated by *Populus* L., *Salix Rosmarinifolia* L. var *brachypoda* (Traktv. Et Mey) Y. L. Chou, *Salix gordejevii* Y. L. Chang et Skv. Therefore, in this study, sparse forests with NDVI values of 0.4–0.6 and dense forests with NDVI values of 0.6–1 were combined into one category; i.e., forests. The dominant vegetation types within different NDVI classification thresholds are shown in Table 4. This classification allows the assessment of changes that have occurred in wetland vegetation over the past three decades.

Iso-clustering unsupervised classification is a multivariate spatial analysis tool used to classify raster bands. The minimum number of classes is two and there is no maximum number of clusters, but the minimum class size should be 10 times larger than the number of layers in the input raster band [25]. The generated classes were grouped and manually recoded using the Reclassification Spatial Analyzer tool in ArcMap software.

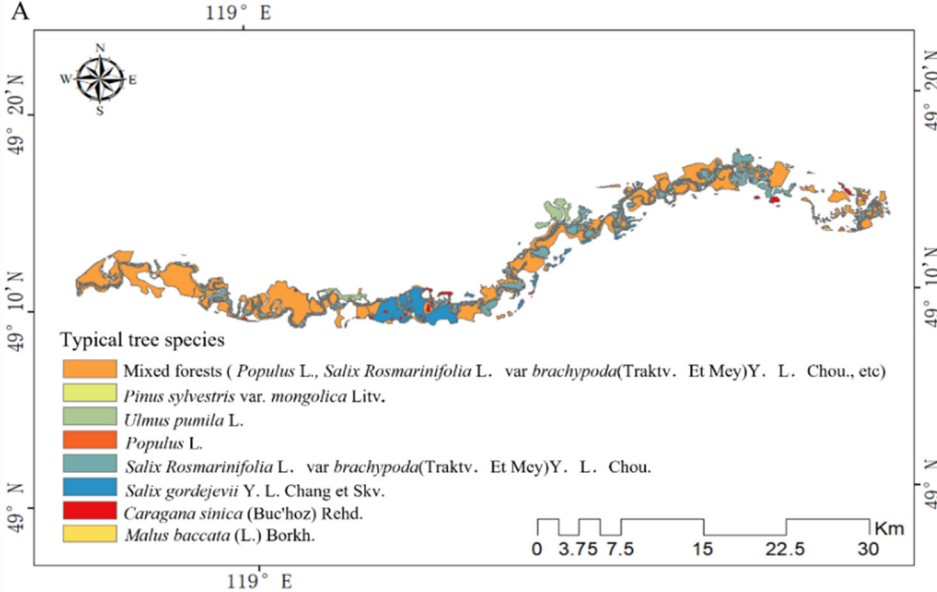

**Figure 2.** *Cont.*

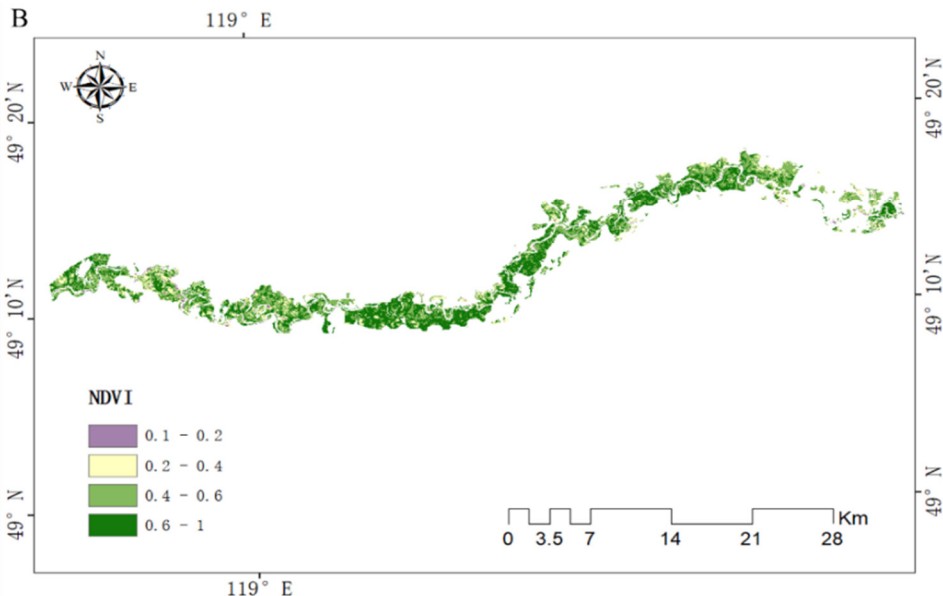

**Figure 2.** The vector diagram of dominant tree species in 2016 and the NDVI of the corresponding year ((**A**) Survey data of typical tree species in 2016 provided by the forestry department; (**B**) NDVI distribution of typical tree species in 2016 calculated based on the survey data provided by the forestry department).

**Table 4.** NDVI classification.

| NDVI | Class | Vegetation Description |
|------|-------|------------------------|
| 0.6–1 | Dense woodland | *Populus* L., *Salix Rosmarinifolia* L. var *brachypoda* (Traktv. Et Mey) Y. L. Chou, *Salix gordejevii* Y. L. Chang et Skv., *Caragana sinica* (Buc'hoz) Rehder, *Pinus sylvestris* var. *mongolica* Litv., mixed forests, etc. |
| 0.4–0.6 | Sparse woodland | *Populus* L., *Salix Rosmarinifolia* L. var *brachypoda* (Traktv. Et Mey) Y. L. Chou, *Salix gordejevii* Y. L. Chang et Skv. |
| 0.2–0.4 | Grassland | *Filifolium sibiricum* (L.) Kitam., *Stipa baicalensis* Roshev., *Leymus chinensis* (Trin.) Tzvel., *Potentilla chinensis* Ser., etc. |
| 0.1–0.2 | Open sand habitat with sparse vegetation | Soil covered with sparse weeds |
| −1–0.1 | No vegetation | Water |

### 2.2.3. Change Detection

Digital change detection is a method of assessing differences in vegetation in different time periods. It has the ability to quantify temporal effects using multispectral image data. There are several methods for accomplishing digital change detection by comparing two or more images of a study site from different periods. Examples are inter-pixel comparison and classification comparison [26]. In this study, three change scenes based on 2019 data in relation to 1987, 2001, and 2010 data were compared using a classification method (the time of data acquisition for this study started in 1987, and the summer 2001 image was chosen as representative because there were no suitable classification images from the summer of 2000). In this study, a spatial model was developed based on Matlab for the analysis and comparison of NDVI change. The modeling process is shown in the flowchart in Figure 3.

Landsat images cannot distinguish between different types of grassland and forest vegetation. In the four time points analyzed in this study (1987, 2001, 2010, 2019), the rainfall in the lower Hailar region was not sufficient to flood the swampy wetlands. Therefore, swampy wetlands are counted among the grassland vegetation types in this study. Meanwhile, the different types of forest vegetation were classified. There are expanding

sandy areas on both sides of the Hailar River [27], namely, the Hailar Sands. Through field research of the sandy places surrounding the study site, the open sand habitat with sparse vegetation in this study was considered to include sandy areas without any vegetation or with sparse vegetation.

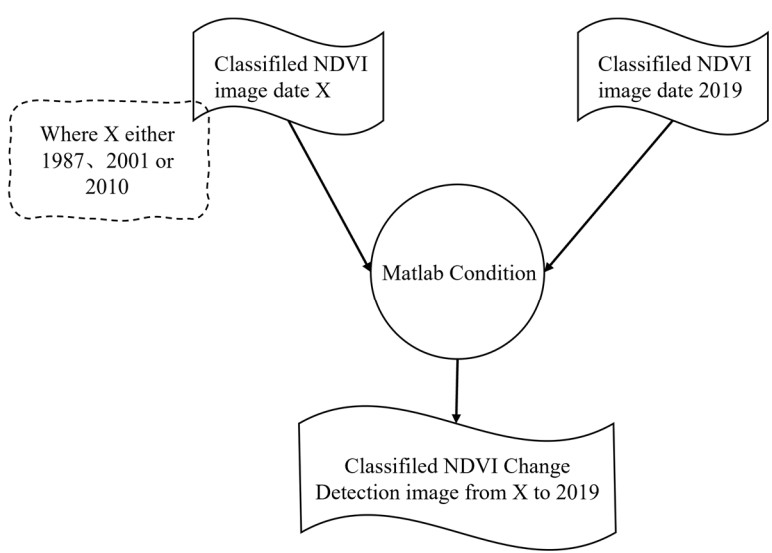

**Figure 3.** Flow chart.

## 3. Results

### 3.1. Accuracy Verification

Table 5 shows the overall classification accuracy assessment and kappa statistics for the NDVI results for 1987, 2001, 2010, and 2019.

Vegetation cover change detection based on the NDVI vegetation index is applicable to most wetlands with >40% vegetation cover [28]. Comparison of past and present Landsat images helps to understand the latest status of wetlands, which can provide a basis for environmental decision-making and management.

**Table 5.** Accuracy assessment and kappa statistics of multi-temporal data from NDVI classification images.

| Time | NDVI | |
|------|------|------|
| | **Classification Accuracy** | **Kappa Statistic** |
| 1987 | 92.10% | 0.884 |
| 2001 | 91.50% | 0.875 |
| 2010 | 93.50% | 0.812 |
| 2019 | 91.00% | 0.911 |

*3.2. Vegetation Change and Spatial Distribution*

This study mapped the vegetation cover status of wetlands in the lower Hailar River section in 1987, 2001, 2010, and 2019 to analyze the patterns of change that have occurred over the past three decades. Figure 4 shows the NDVI classification maps for the summer vegetation growth spurt at four time points in 1987, 2001, 2010, and 2019. Table 6 shows the areas of bare soil, grassland, and forest in the study site (in km$^2$ and as percentages of the total area, based on NDVI).

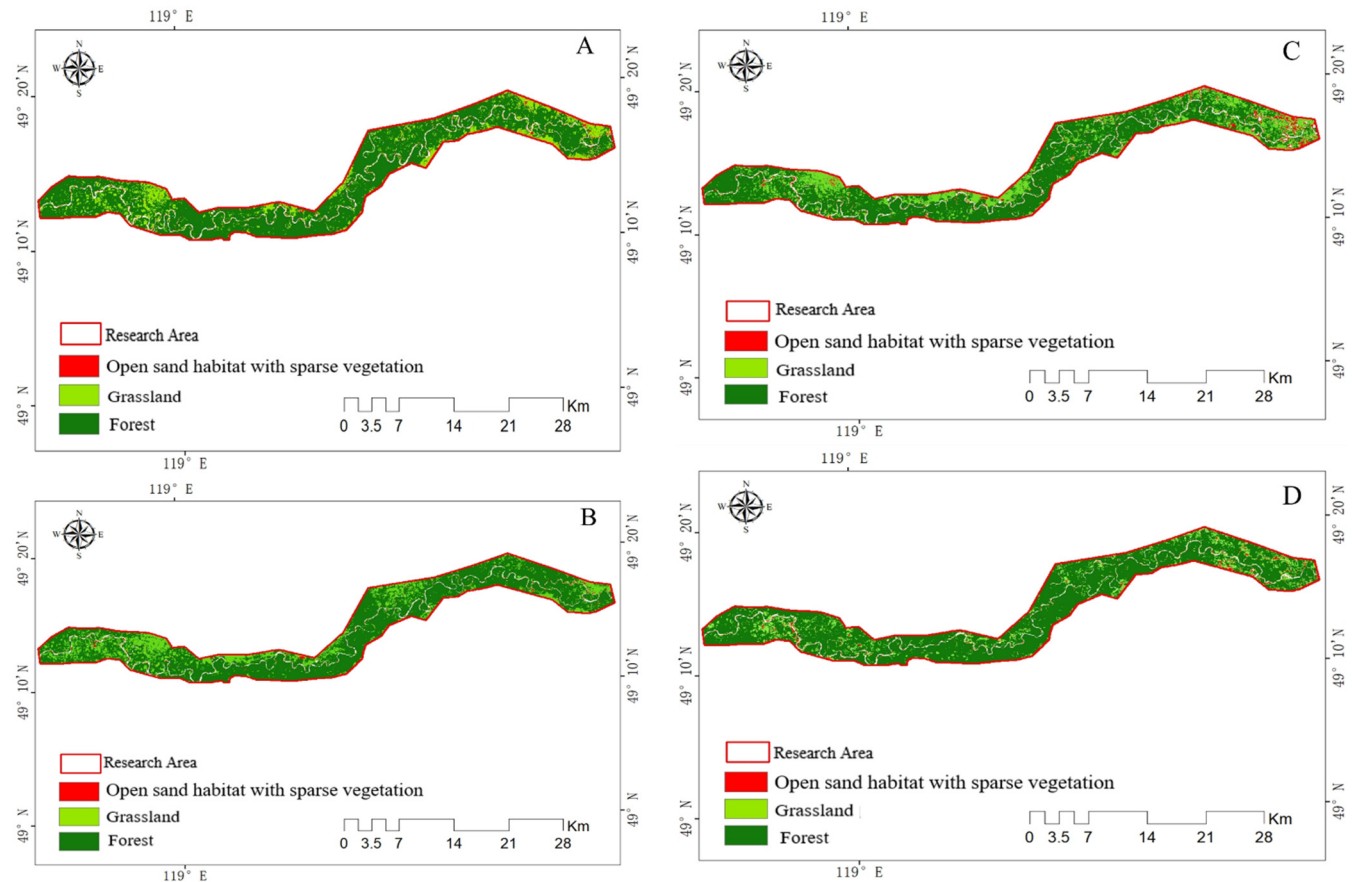

**Figure 4.** NDVI maps of the Hailar River from (**A**) 1987, (**B**) 2001, (**C**) 2010, and (**D**) 2019.

**Table 6.** Areas of the study site (km$^2$ and percentages) based on NDVI classes.

| Code | Class | 1987 | | 2001 | | 2010 | | 2019 | |
|---|---|---|---|---|---|---|---|---|---|
| | | Km$^2$ | % | km$^2$ | % | km$^2$ | % | km$^2$ | % |
| 1 | Open sand habitat with sparse vegetation | 4.96 | 1.57 | 4.22 | 1.34 | 11.32 | 3.58 | 6.04 | 1.91 |
| 2 | Grassland | 55.79 | 17.66 | 70.85 | 22.42 | 82.98 | 26.26 | 42.62 | 13.49 |
| 3 | Forest | 255.21 | 80.77 | 244.30 | 77.32 | 221.65 | 70.15 | 271.12 | 85.81 |
| Total | | 315.95 | 100 | 319.37 | 100 | 315.96 | 100 | 319.78 | 100 |

The detailed vegetation status of the wetlands in the upper section of the Hailar River is shown below. The total area of all categories shows fluctuating changes in the wetland area during the study period. Over the past 30 years, vegetation cover (grassland, woodland) accounted for about 311.13 km$^2$ or 97.91% of the total land cover of the entire study site.

Open sand habitat with sparse vegetation: During the study period, open sand habitat with sparse vegetation in the study site decreased, then increased, and then decreased

again. In 1987, the area of open sand habitat with sparse vegetation was 4.96 km$^2$ (1.57%); by 2019, its coverage increased to 6.04 km$^2$ (1.91%), an increase of 1.08 km$^2$ at an average rate of 0.033 km$^2$ (0.34%) per year. At the four time points, the largest area of open sand habitat with sparse vegetation occurred in the summer of 2010 and was 11.32 km$^2$.

Grassland: During the study period, this category showed a more significant change than the other two. Grassland change can be divided into two intervals: an increase followed by a decrease. Between 1987 and 2010, the total area of the grassland category increased from 55.79 km$^2$ (17.66%) in 1987 to 82.65 km$^2$ (26.26%) in 2010, a total increase of 27.19 km$^2$ over this time period at an average rate of 1.24 km$^2$ per year. Between 2010 and 2019, the grassland area decreased from 82.98 km$^2$ to 42.62 km$^2$, a total decrease of 40.36 km$^2$ at an average rate of 4.04 km$^2$ per year. The total decrease in grassland area during 1987–2019 was 13.17 km$^2$.

Forest: During the study period, this category showed a clearly opposite trend to that of grassland. Forest change can likewise be divided into two intervals of change: a decrease followed by an increase. Between 1987 and 2010, the total area of forest decreased from 255.21 km$^2$ (80.77%) in 1987 to 221.65 km$^2$ (70.15%) in 2010, a total decrease of 33.56 km$^2$ at an average rate of 1.53 km$^2$ per year. Between 2010 and 2019, the forest area increased from 221.65 km$^2$ to 271.12 km$^2$, a total increase of 49.47 km$^2$ at an average rate of 4.95 km$^2$ per year. There was an overall increase in the area of forest of 15.91 km$^2$ from 1987–2019.

*3.3. Vegetation Change Inspection*

Vegetation changes in the different NDVI categories were determined relative to the state in summer 2019; hence, for the periods 1987–2019, 2001–2019, and 2010–2019.

As shown in Figure 5 and Table 7, from 1987–2019, the total open sand habitat with sparse vegetation area converted to vegetation (forest and grassland) was 39.3 km$^2$. At the same time, the area converted from vegetation to open sand habitat with sparse vegetation was 39.76 km$^2$; so, in this time period, the area of open sand habitat with sparse vegetation increased by 0.46 km$^2$. From 2001–2019, the total open sand habitat with sparse vegetation area converted to vegetation (forest and grassland) was 27.07 km$^2$. Meanwhile, the area of vegetation converted to an open sand habitat with sparse vegetation was 37.15 km$^2$. So, in this time period, the area of open sand habitat with sparse vegetation increased by 10.08 km$^2$. From 2010–2019, the total area of open sand habitat with sparse vegetation converted to vegetation (forest and grassland) was 86.56 km$^2$; meanwhile, the area of vegetation converted to open sand habitat with sparse vegetation was 24.94 km$^2$. Within this time period, the area of bare soil decreased by 61.65 km$^2$ and, in the three time periods, a large area of grassland was converted to forest and a small area of forest was converted to grassland.

**Table 7.** Changes in the area occurring in the three time periods.

| Code | Change Type | 1987–2019 | | 2001–2019 | | 2010–2019 | |
|---|---|---|---|---|---|---|---|
| | | Area (km$^2$) | (%) | Area (km$^2$) | (%) | Area (km$^2$) | (%) |
| 12 | Open sand habitat with sparse vegetation to Grassland | 16.8 | 2.69 | 17.99 | 2.65 | 48.92 | 6.22 |
| 13 | Open sand habitat with sparse vegetation to Forest | 22.5 | 3.6 | 9.08 | 1.34 | 37.64 | 4.79 |
| 21 | Grassland to Open sand habitat with sparse vegetation | 17.43 | 2.79 | 18.27 | 2.69 | 15.28 | 1.94 |
| 23 | Grassland to Forest | 345.49 | 55.28 | 448.83 | 66.05 | 563.63 | 71.68 |
| 31 | Forest to Open sand habitat with sparse vegetation | 22.33 | 3.57 | 18.88 | 2.78 | 9.63 | 1.22 |
| 32 | Forest to Grassland | 200.46 | 32.07 | 166.52 | 24.5 | 111.16 | 14.14 |
| Total | | 625.01 | 100 | 679.57 | 100 | 786.25 | 100 |

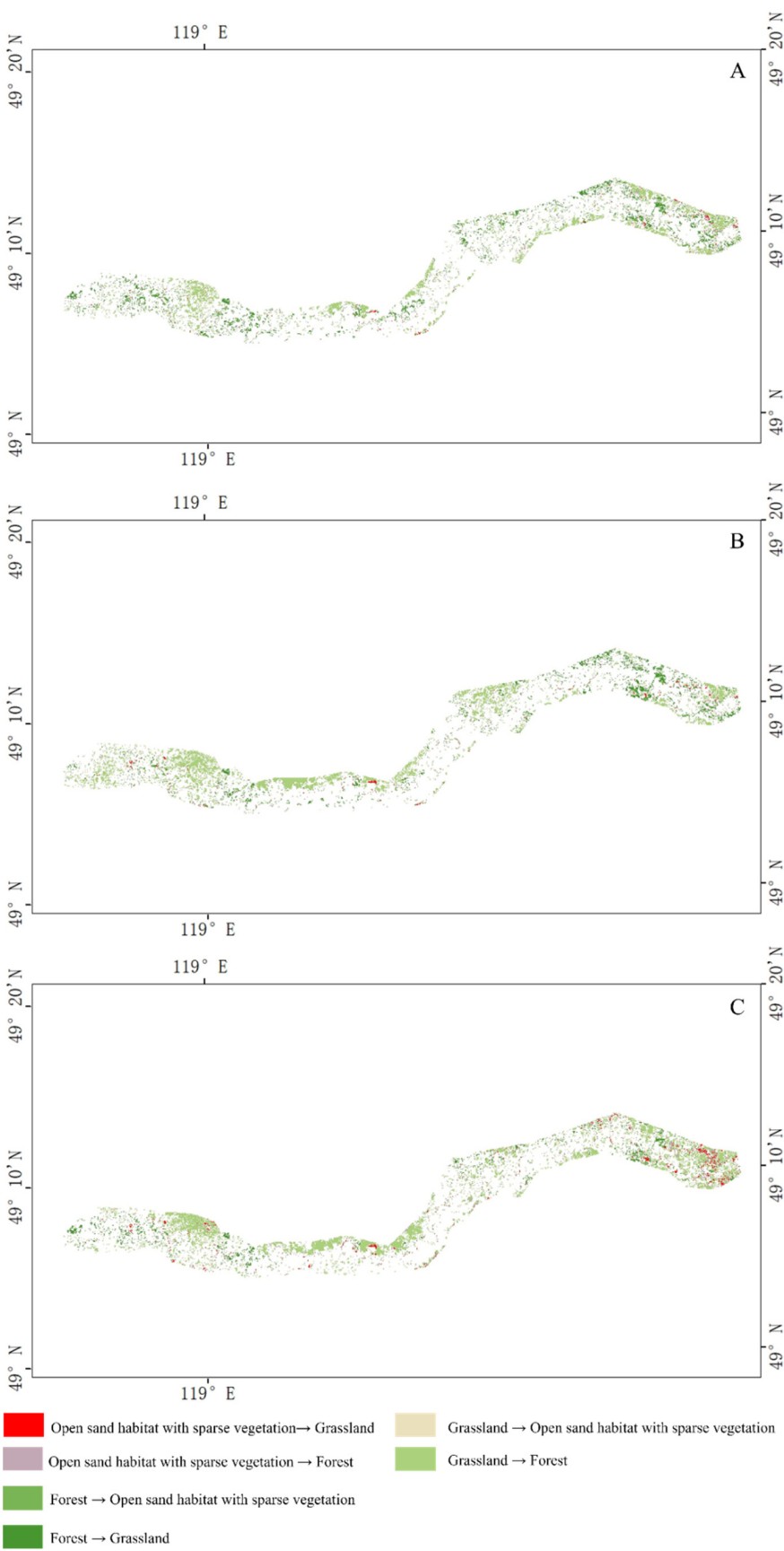

**Figure 5.** NDVI category changes occurring in the three time periods of (**A**) 1987–2019, (**B**) 2001–2019, and (**C**) 2010–2019.

## 4. Discussion

### 4.1. Vegetation Change Digital Model Evaluation

Remote sensing technology provides an accurate, rapid, and cost-effective method for wetland change detection [11]. Various digital methods can be used to detect changes by comparing two or more images of a study site from different periods, such as pixel-to-pixel comparison before and after vegetation change [26]. The classification change detection method is one of the most appropriate and commonly used change detection techniques. This technique can easily provide a change matrix from which transfers between one vegetation type and another can be visualized. This method identifies areas of change as pixel-by-pixel differences between categories after obtaining a classification image [29]. This can clearly show areas of change as well as transformation categories. The model proposed in this study estimates the process of vegetation change by studying different periods.

In this long-term study, based on Landsat data of the summer growth season in 2019, we detected change at three time points in 1987, 2001, and 2010. We generated a digital spatial model of vegetation change through Matlab programming and used it to detect changes in an NDVI map. The model proposed in this study has certain advantages in quantifying vegetation change in the floodplain wetlands. This evaluation method based on remote sensing imagery and Matlab modeling has good operability and detection accuracy. It is not only suitable for research on vegetation diversity in submerged areas but also provides a valuable reference for river management decision-makers pursuing wetland protection and other aspects.

### 4.2. Reasons for the Differences in the Spatial Distribution of Vegetation

The degree of connection between the main channel of a river and a riverine wetland will affect riverine wetland vegetation [30]. It was found that, in the natural state, the composition and structure of plant communities, and their changes, are closely related to flooding occurrence [31].

The overall high NDVI values observed near riverbanks indicate that most of the vegetation near these locations benefits from water recharge from the Hailar River. The fertile silt left after periodic flooding also improves the nutrient content of the soil on both banks, and water vapor improves the surrounding microclimate, thus forming a high-biomass plant belt along the riverbanks. Even plants of sandy open sand habitat with sparse vegetation (*Caragana microphylla* Lam, *Cleistogenes squarrosa* (Trin.) Keng, etc.) form high biomass and high canopy closure plant communities near riverbanks, which greatly weaken the mobility of sand dunes. In the horizontal direction, the vegetation type varied with distance from the riverbank. In Figure 6, the riverbank vegetation types, in the direction from the river to the habitats more distant from the river, are forest (*Populus* L., *Salix Rosmarinifolia* L. var *brachypoda* (Traktv. Et Mey) Y. L. Chou, *Salix gordejevii* Y. L. Chang et Skv., *Caragana sinica* (Buc'hoz) Rehder, *Pinus sylvestris* var. mongolica Litv., mixed forests, etc.), grassland mixed with forest, and grassland. As shown in Table 4, the NDVI thresholds for grassland are lower than those of forest, so the NDVI tends to decrease with distance from the riverbanks. Due to the special environment of riverine wetland systems, wetland vegetation development is associated with the river. The results show that the direct impact of the river on the vegetation on both sides of the river decreases with distance. If river runoff decreases and flood cycles become longer, it will adversely affect the wetland forest and riverine meadow ecosystems that depend on periodic flooding, leading to degradation of wetland forest areas, reductions in near-riparian wetland grassland areas, and degradation of wetland grassland systems [32,33].

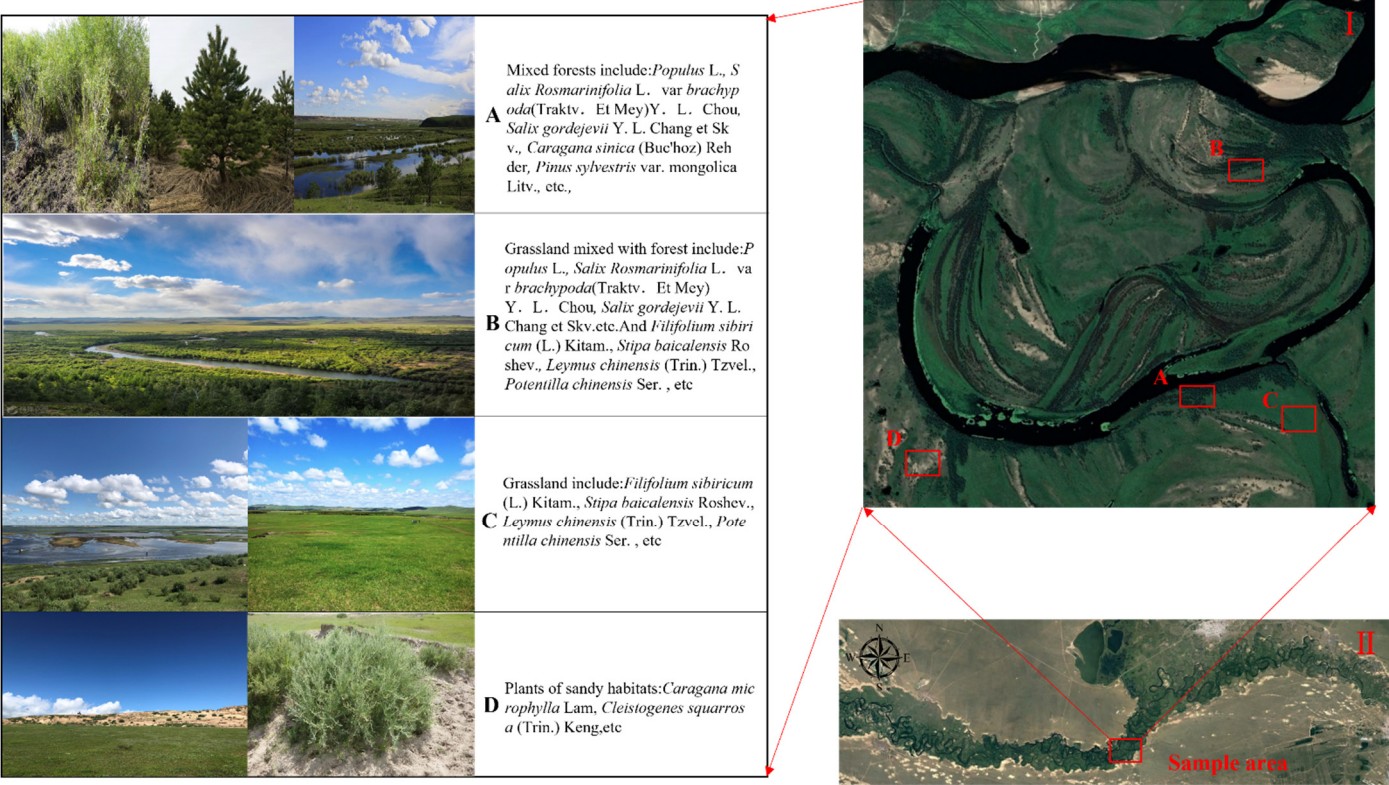

**Figure 6.** The description of vegetation types in the study area.(I: Is an enlarged view of a typical area in the picture II; II: It is a typical area with well-developed river floodplain wetland vegetation in the study area).

### *4.3. Vegetation Change Process and Its Influences*

Figure 7 shows the changes in NDVI categories occurring in the three time periods 1987–2019, 2001–2019, and 2010–2019. Figure 7A shows that of the first three transformation categories, the changes occurring between 2010–2019 are obvious, while the conversion of forest to open sand habitat with sparse vegetation and grassland changed significantly between 1987–2019. Figure 7B shows that within the three time periods, among the six transformation categories, the conversion of grassland to forest changed most significantly, followed by conversion of forest to grassland, indicating that more pronounced vegetation changes occurred within the three time periods.

Vegetation change in floodplain wetlands is affected by many man-made and natural factors, such as water supply, rainfall, and runoff [34–36]. The environment in the study site of the lower Hailar River is complex and diverse with a variety of ecosystems, such as swampy wetlands, river meadows, scrub, and forest. This area is a transition area between the river system and the surrounding environment and is an important ecological transition zone in this watershed, which is the most active part of the ecosystem in terms of energy and material transfer, and transformation [37,38] and is very sensitive to external changes. These ecosystems are more dependent on runoff recharge from the Hailar River and periodic flood recharge, and their vegetation trends have both positive and negative possibilities. In areas with sufficient and stable water recharge, the vegetation gradually develops for the better, herbaceous plant communities gradually evolve into scrub woodland plant communities, bare soil areas remain stable, and biodiversity continues to grow. For areas lacking in water recharge, the speed of land desertification may exceed the speed of vegetation recovery, due to the influence of the adjacent.

Hulunbuir Sandy Land, thus forming a vicious cycle and leading to reverse ecosystem changes, such as vegetation changes to open sand habitat with sparse vegetation.

Human activities also affect vegetation changes. In this study, statistics of the Hailar Region for 1987, 1995, and 2019 were collected from Statistical Yearbooks. Table 8 shows that when regional GDP increases at a relatively fast rate (more than 10 times that of the previous period), the population growth rate slows down, the growth rate of industrial production slows down, the growth rate of food production declines, and the growth rate of arable land increases by 266.99%. The increase in population and economic output leads to an increase in the scale of industrial and domestic water consumption in the river basin, resulting in an increase in the intensity of water resources development. These human development activities interact with the surrounding ecosystems and have become the main driving force in land expansion, agricultural intensification, and water resource exploitation in the Hailar area, and have profound impacts on the vegetation of floodplain wetlands.

In recent years, Chinese national and local governments have issued a series of wetland vegetation protection projects. For example, in 2003, the Chinese government approved the 2002–2030 National Wetland Protection Plan, which aims to restore natural wetlands and establish nature reserves [39]. In 2011, Russia and China adopted the Russia-China Amur River Basin Transboundary Protection Zone Development Strategy to 2020, which announced that the protection of wetlands would be a top priority [40]. Through a series of human interventions, wetlands have been restored and their vegetation protected. Therefore, in order to protect and restore wetland ecosystems, it is still necessary to sustainably manage natural wetlands [39].

This research suggests that in the process of rapid urbanization, industrialization, and agricultural modernization, human beings should take greater responsibility for protecting well-developed wetlands and prevent their degradation. This requires, for example, better management of farmland irrigation, wetland development, and water resources.

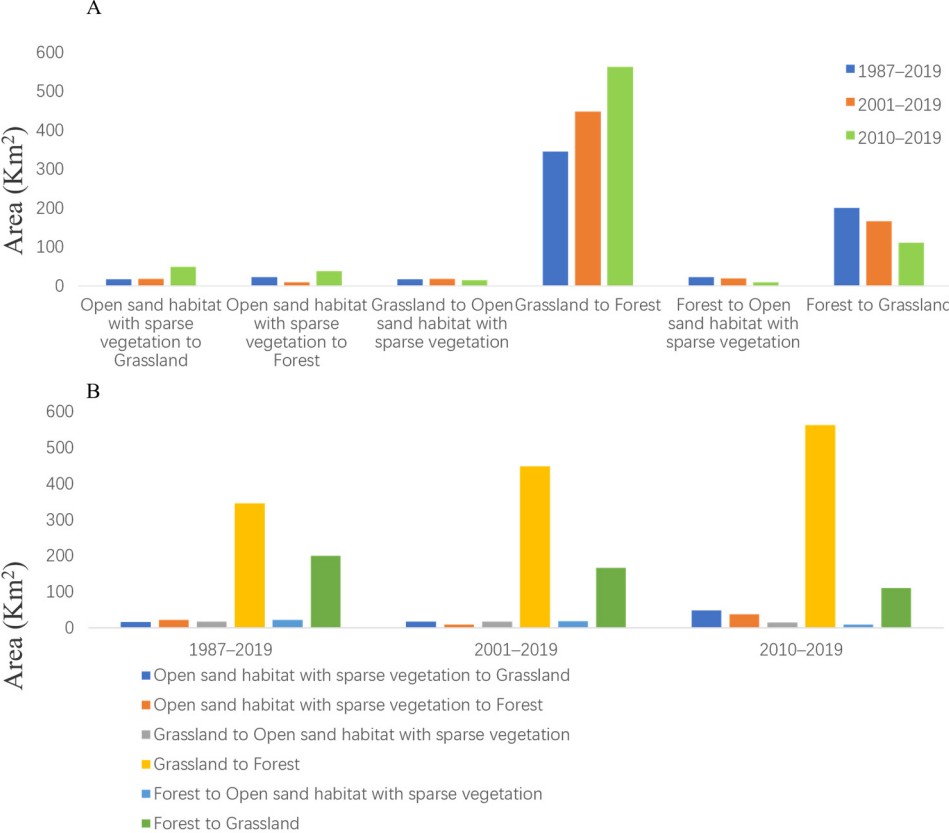

**Figure 7.** Comparison of vegetation changes within different periods ((**A**) the changes of the three categories in three different time periods; (**B**) the changes of the 6 conversion categories in the same time.).

**Table 8.** Social and economic indicators reflecting human activity in the Hailar River area in 1987, 1995, and 2019.

| Indicator | 1987 | 1995 | 2019 | 1987–1995 Rate of Change (%) | 1996–2019 Rate of Change (%) |
|---|---|---|---|---|---|
| Gross regional product (10,000s of yuan) | 289,664 | 994,286 | 11,930,295 | 243.25 | 1099.89 |
| Population (10,000s) | 12.5 | 22.21 | 28.74 | 77.68 | 29.40 |
| Total industrial output (10,000s of yuan) | 24,652 | 80,986 | 109,068.1 | 228.52 | 34.68 |
| Output of main foodstuffs (tons) | 8720 | 44,778 | 87,456 | 413.51 | 95.31 |
| Cultivated land area (km$^2$) | 188 | 242.13 | 888.6 | 28.79 | 266.99 |

## 5. Conclusions

Remote sensing technology provides an accurate, fast, and economical method of vegetation change detection. It solves the difficulty and time-consuming implementation of traditional techniques. In this study, vegetation classification maps were produced by an unsupervised classification tool and comparative calculations of change in coverage of vegetation classes were conducted. A vegetation change detection model based on Matlab was used to analyze the changes between different categories. The main conclusions are:

(1) The wetlands in the lower Hailar River study site underwent significant changes in vegetation between 1987 and 2019. Open sand habitat with sparse vegetation areas increased by 1.08 km$^2$; grassland areas decreased by 13.17 km$^2$ and forest areas increased by 15.91 km$^2$.

(2) The spatial distribution of NDVI values differed across the study site. The NDVI was high overall and the vegetation types varied with distance from the riverbank. NDVI tended to decrease with distance from the riverbanks and the main vegetation types were, from near to far, forest (*Populus* L., *Salix Rosmarinifolia* L. var *brachypoda* (Traktv. Et Mey) Y. L. Chou, *Salix gordejevii* Y. L. Chang et Skv., *Caragana sinica* (Buc'hoz) Rehder, *Pinus sylvestris* var. mongolica Litv., mixed forests, etc.), grassland mixed with forest, and grassland.

(3) The ecological environment of the lower Hailar River study site is complex and diverse. It contains a variety of ecosystems such as swampy wetland, river meadow, scrub, and woodland. All of these ecosystems are dependent on runoff recharge from the Hailar River and periodic flood recharge. There are two possibilities for positive and negative vegetation change trends. In areas where the water supply is sufficient and relatively stable, the cover of forest vegetation was gradually increasing and the herbaceous plant community gradually evolves to scrub woodland. In areas where the water supply is lacking, there are changes in sense of decrease of forest vegetation and increase of open sand habitat with sparse vegetation.

(4) This study suggests that in the process of rapid urbanization, industrialization, and agricultural modernization, humans should take greater responsibility for protecting well-developed wetlands and, at the same time, better manage farmland irrigation, wetland development, and water resources.

**Author Contributions:** Conceptualization, X.D. and Z.C.; methodology, X.D.; software, X.D.; validation, X.D. and Z.C.; formal analysis, X.D..; investigation, X.D.; resources, X.D.; data curation, X.D.; writing—original draft preparation, X.D.; writing—review and editing, X.D.; visualization, X.D..; supervision, Z.C.; project administration, Z.C.; funding acquisition, Z.C. Both authors have read and agreed to the published version of the manuscript.

**Funding:** This research received the Fundamental Research Funds for the Central Universities (Grant No. 2018BLRD18).

**Data Availability Statement:** Data is not available.

**Conflicts of Interest:** The authors declare no conflict of interest.

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
