# Peer review of "Digital Examination of Vegetation Changes in River Floodplain Wetlands Based on Remote Sensing Images: A Case Study Based on the Downstream Section of Hailar River"

_forests, doi:10.3390/f12091206_

Round 1
Reviewer 1 Report
The authors reponded all comments and made a careful, almost complete and almost satisfying revision. However, I still found one major problem. This is about the usage of a basic scientific term „Habitat” that has been used here incorrectly. I strongly suggest to correct this name. Please, correct the whole ms.: i.e. correct all sentences, all figures and all tables where you referred open sand areas only as „Habitats”. You should specify what habitat you mean:
Suggested versions:
„open habitat” or „open sand habitat” or „open sand habitat with sparse vegetation” (all would be correct but using only „habitat” is not correct).
The name „Habitat” was introduced only in the new revised version for describing areas that were called as „open soil” in the previous manuscript version. Authors now use „Habitat” to depict one specific vegetation type (for open sand areas with sparse weed vegetation). However, this is not correct. The term „habitat” is a general term in ecology and in vegetation science and it cannot be used for only one vegetation type. In fact the authors studied here the patterns of several different habitats: they had „forest habitat”, „grassland habitat” and „open habitat”. Consequently and logcally the word „habitat” cannot be used to describe one of these habitat types without further characterization.
There is huge literature of various habitat types with related definitions and characterizations. For example, please, refer the recent paper by
Chytrý et al.
EUNIS Habitat Classification: Expert system, characteristic species combinations and distribution maps of European habitats. Applied Vegetation Science 2020 pp. 648-675
https://doi.org/10.1111/avsc.12519
Descriptions and names of the vegetation types are central issues in this paper. The present term „Habitat” is incorrect and misleading. Correcting all sentences, all figures and all tables in the manuscript for a proper name is necessary.
Other minor issue:
1, Remote sensing images used in this paper represent four dates (four time points). However in one sentence authors mentioned „four time series” page 5 line 150. Please, correct „four time series” to „four time points”.
2, Page 7 L 197-199: the sentence „Through field research of the sandy places surrounding the study site, the habitat in this study was considered to include sandy areas without any vegetation or with short-lived vegetation” is not clear. Please, reword and clarify what do you mean here.
3, Page 17 L 401: the sentence „The code is already in the supplementary material” is not clear. Please, reword and clarify what do you by „codes” here. (do you mean internet site or particular data or some keys of the complete data set?)
4, Table 9. 8720 should be 8,720 and 44778 should be 44,778 ( , was missing).
Columns titles: 1987-1195(%) and 1996-2019(%) it is not clear what (%) refers.
Except these minor typos and the major problem of the improper usage of term „Habitat”, all other concerns and suggestions made by referees were perfectly considered and the manuscript become much better.
Reviewer 2 Report
See the attachment – zip file with the review report and commented PDF of the paper.

Author Response
请参阅附件。

Round 2
Reviewer 1 Report
I have checked the new version. The authors made a careful, satisfying revision. In my opinion, the paper can be accepted now.
I found few very small points for further amendmends. However, these are small typos that can be corrected during proof-reading:
Page 2 L 98 asperata should be in italic
Page 5 L 173 a space is missing after bachypoda
Page 13 L 339 water vapour should be water vapor
Page 17 L 445 a space is missing after bachypoda
This manuscript is a resubmission of an earlier submission. The following is a list of the peer review reports and author responses from that submission.
Round 1
Reviewer 1 Report
Review of the manuscript entitled „Digital examination of vegetation succession in riverine wet-lands based on remote sensing images: -- A case study based on the downstream section of Haila River“ submitted by X. Dong and Z. Chen to the journal Forests
Overall evaluation
The paper fits well into the scope of the journal Forests, presenting new and original results. The most valuable is the comparison of Landsat images for quite a long time period, indicating important changes in the vegetation cover. What I completely miss is the discussion on underlying processes of these changes. Overall elaboration of the manuscript (language, clarity, etc.), however, shows many serious shortcomings. The whole paper must be substantially improved.
Please, note that below I list only the major comments, at first the general ones, appearing in various parts of the manuscript, and then the comments specific to individual parts of the paper. Please, note that small but important comments are also included directly in the PDF of the manuscript.
General comments
The manuscript is not very well prepared, in particular, there are following problems:
- The text urgently needs linguistic correction, as some of the sentences are very long and difficult to understand. Some of the sentences even do not make any sense, as there are missing words or, in contrast, some parts are doubled. Some phrases are very unusual, e.g. “ecological environment”, “wildlife ecosystems”, or are like an accidental grouping of words, e.g. “ecological security maintenance area for biodiversity maintenance”. Some of the sentence constructions are too complicated and the same thing could be expressed by substantially lower amounts of words; in some cases, I suggested some words for deleting (see the attached PDF of the manuscript with my suggestions).
- A careful check of the references is necessary. Probably all of the references are incorrect; see my comments in the PDF of the manuscript. I think that the authors used some automatic system which, actually it is recommended in the author guidelines. However, they always should check the reference list manually! Unfortunately, their system did not work well – for instance, the words from the journal names are incorporated into the author lists, journal names are incomplete, volume and pages are sometimes missing, etc.
- As far as I know, the authors should already in the submitted version of the manuscript transform all the citation in the text into the numbers, according to the order of each citation in the text. Here the authors numbered the citations on the reference list, but their order is not according to the order in the text. The order in the text is ignored and an alphabetical order is followed. In the text, the author names are kept instead of being transformed into the numbers.
- All the scientific (Latin) names should be written in italics (with the exception of “var.” and similar parts of the names).
- The structure of the paper does not follow the guidelines, see the parts 2. and 3., which should be put together and hierarchically under “Materials and Methods” and then divided accordingly on sub-chapters.
Specific comments
Various parts of the text
- What is actually the name of the river studied? Altogether three forms occur in the text: Haila (e.g. the title), Hailar (e.g. abstract) and Hela (e.g. abstract). It is necessary to select only a single form and unify the text accordingly.
- Succession = vegetation change during a certain time period, usually leading to a higher level of vegetation complexity. So, it is not correct to use the term “vegetation succession change” (e.g. in introduction) if you speak just about the “vegetation change” as I understood from the paper. Of course, succession can be changes as well, e.g. its direction or speed, based on the other environmental parameters.
Specific parts of the text
Introduction
– p. 2, 3rd paragraph – the authors state that “but few studies have been conducted for the riverine wetlands in the lower section of the Hailar River”. Please, could you provide the references for the already existing papers?
2.1. Study area
– p. 2 – what the authors mean with the term “good vegetation”? Is it “well developed vegetation”? Or something else?
– “primeval forests” – may be the term “pristine forests” would be more appropriate?
3.2. Vegetation classification
- the very long sentence on p. 5, 1st paragraph is very difficult to follow, it is necessary to divide it on several shorter sentences. I also do not understand why in the end of the sentence is written about three vegetation types, although in the table 4 and in the beginning of the sentence there are listed five vegetation types. It is unclear, if the three vegetation types were mention in some of the cited publication or are really related to “this study” (i.e. the reviewed manuscript)
– Figure 2 – it is necessary to list all the tree or shrub species under their Latin names – now there are some Latin names, some English names and even a Chinese name. For the mixed forests, it would be useful to list up to three dominant tree species into the parentheses at the figure or elsewhere in the text. In my opinion, it is also rather difficult to follow the two types of information in one and the same figure. There are colours of vegetation types and colours of the NDVI. I’m really do familiar with common practise, as I do not work with this type of mapping but it seems that there are overlaps of some colours and thus probably some information is lost. It applies particularly for the parts which are marked as sparse or dense woodlands – one would expect that there would be mapped the dominant species but they are not visible due to the green NDVI colours. If the two types of information should be in the single figure, it would be necessary to present the NDVI like some type of section lining.
– Table 4 – similarly as at Figure 2, it is necessary to use Latin names at all the species (for genera, e.g. poplar, it is possible to write e.g. “Populus spp.”, of more species of the genus occur in the study
area). I’m also unsure if „linophyllum“ is a part of the name of some species (i.e. genus name was accidentally deleted) or it should be a genus (however, I only found a genus Lenophyllum on internet).
4. Results and Discusssion
4.1 Accuracy verification
– some sentence and reference to the kappa statistics (see Table 5) would be useful to incorporate into the methods
4.2. Vegetation change and spatial distribution
– the Figures 4A–4D should be of the same size like the figure 2. Now the details in the figures 4A–4D are poorly visible, after zooming in the PDF it seems that the resolution is insufficient. The same applies for the Figure 5 in the next sub-chapter.
– p. 8, evaluation of increase and decrease of individual vegetation types – I think that it does not make too much sense to include average rate of the decrease/increase per year. From the data it is visible that the changes do not follow any rules, they seem to be highly irregular. Moreover, the interval between the first (1987) and the second (2001) recording of the Landsat images is longer than between the second and third (2010), and third and fourth (2019) recording.
– The last two sentences of this section are very important, as they give some more details on the changes, however, the description includes many unclear points (here marked in yellow and commented below): “Even sandy plants formed a high biomass and high depression plant community near the river bank, which greatly weakened the mobility of the sandy land; along the vertical direction, the vegetation type varied with the distance of the river bank. NDVI tends to decrease from near to far, and the vegetation types of river banks from near to far are scrub forest (poplar, marsh willow, yellow willow, mallow, camphor pine, mixed forest), mixed grassland and grassland in order, and the direct impact of rivers on vegetation on both banks decreases with distance, and if river runoff decreases and flood cycles become longer, it will adversely affect swampy wetlands and river meadow ecosystems that depend on periodic flooding, causing Wetland degradation, reduction of near riparian wetland meadow area, and degradation of wetland grassland systems(Karimi, Saintilan, Wen, & Valavi, 2019; Wen et al., 2013)”.
– What are the “sandy plants”, do the authors mean “plants of sandy habitats”?
– “high depression plant community” – what do you mean with this? And is it the community od “sandy plants” or a different community?
– “sandy land” – may be better to write “sand dunes”, if the places can be considered for dunes, or “sandy sites” or “sandy places”
– “NDVI tends to decrease from near to far” – Do you mean “NDVI tends to decrease in the direction from the river banks to the sites more distant from the river”?
– “mixed grassland” – it appears for the first time now. How the mixed grassland looks like? Is it mixed with forest? Or is it a mixture of two or more grassland types? Please, specify!
– “grassland in order” – I do not understand. Can you delete “in order”?
4.3. Vegetation change inspection
– p. 9, 2nd paragraph – the authors mention several habitat/vegetation types which are not mentioned in the main results (figures and tables). It is clear, that Landsat images probably do not enable to differentiate among the various types of grasslands. However, what about the swampy wetlands? Are they hidden under the tree canopy and thus not included into the mapping? Or are they included into the category “open soil”? Is the open soil actually free of any vegetation, or is it a habitat of temporarily flooded and exposed lower river banks with specific short-lived vegetation? Please, add some explanatory sentence into the method!
– p. 9, 2nd paragraph – What do you mean with the positive and negative possibilities of succession? And what the statement “vegetation gradually develops for the better” means? Do you consider the forest vegetation to be “better” than the grassland or open soil? Please, describe clearly – e.g. development from non-forest to forest vegetation types, etc. I would also like to know, what is “vicious” on the cycle of succession (and natural disturbances – would be worth mentioning in your paper!) at sandy places. Moving sands/sand dunes and the processes repeated disturbances and succession are natural processes and I’m afraid that all the above statements are somehow personified and such a writing style is not proper for the ecological paper.
– p. 10–11 – content of the Table 7 and Figure 6 is overlapping; I recommend deleting of the Figure 6.
5. Conclusions
Point (4) of the conclusions is interesting but there is none relationship of this point to the data presented. I would expect that the underlying processes of the vegetation changes will be discussed, at least in some coarse categories like anthropogenic impact versus natural processes (accumulation and denudation in the river floodplain). If the details of these processes are unknown in this particular case, it should – be stated clearly and accompanied by the discussion on studies from similar geographic conditions. Such a discussion would also legitimaze the point 4 of the Conclusions.
References – must be checked and completed.

Reviewer 2 Report
The study assessed vegetation changes from a series of Landsat satellite images. Percentage changes and transformations of three major riverine wetland habitat types (open soil, grassland, forest) were estimated.
Strengths of the study:
1, This is a long-term study assessing 30 years of vegetation changes.
2, Results seems to be reliable because simple vegetation maps were estimated using robust, standard methodology.
3, Trends were explored with novel, interesting, retrospective scaling when a recent vegetation map has been compared to previous maps with increasing time intervals between the two estimates.
4, Understanding vegetation states and transformations in wetlands is an important topic and the methodology used here has the potential to provide valuable information for local authorities.
Weaknesses of the study:
1, Introduction is too general. Research background (similar studies about riverine vegetation dynamics) has not been reviewed. Similar results from other studies were not discussed.
2, Aims 1st is identical with aim 2nd while aim 3rd is poorly elaborated (spatial data, i.e. vegetation maps have been developed but without quantitative analyses and explicit results).
3, Causal background – causes of vegetation state transformations - is missing or superficially treated. Some potential drivers (runoff recharge and periodic floods) have been mentioned but without quantitative data or analyses.
4, Some methods (that were used) have not been mentioned in Methodology section (classification accuracy and Kappa statistics).
5, NDVI thresholds for recognizing certain vegetation types nicely estimated from 2016 data. However, it was not analyzed and discussed whether such thresholds can be used for all images (whether these technical thresholds were invariant over longer periods of time or NDVI photos should be calibrated separately for each year).
6, Links with practical aspects (urbanization, industrialization, agricultural modernization) have been mentioned but these important aspects remained poorly explained.
7, The whole text needs basic revision of English. Specific scientific terms need clear definitions or revisions.
8, There are serious technical problems with citations and with the list of References. I have also concerns about the selection of studies for specific methods or evidences. Please, try to search for original publications and for more relevant examples.
Other comments
(Unfortunately the manuscript did not have line numbering. Therefore, it was very hard to perform a formal review and I could not refer specific sentences. Due to these limitations I had to select and I will focus only on the most important problems.)
1, What is the name of the river? Title shows “Haila”, main text shows “Hailar”, some figure (Fig. 1) shows “Hailaer”.
2, Are you studying vegetation succession or vegetation changes? Succession used to be an ordered series (!) of vegetation changes not only transformation between two vegetation types.
3, Is “open soil” a vegetation type? The term “habitat” could be more appropriate here.
4, Please, characterize the overall state and history of Hailar river. Can this river and its riverine habitats be considered as natural habitats or the river had been regulated and riverine habitats were changed by human? All forests are natural? Can the detected increase of forests be explained by natural dynamics or some forest were planted (man-made) in these habitats.
5, How the total study area (border of vegetation maps) has been delineated? Could you describe the vegetation over the whole catchment area? How did the overall landscape, land use and climate change over the last 30 years?
6, “True color image” on Fig. 1. is not visible (looks like a black and white version).
7, I think Table 2 and 3 could be deleted as they show results from other studies and it was not clear what were the role of these data in the present study.
8, Fig. 2. It is not clear what do you demonstrate here. How the vector map of dominant tree species was overlaid with NDVI values? Original maps (NDVI and Forestry map of tree species) should be shown as well.
9, You defined three vegetation types for mapping. However, Table 4 shows thresholds for five vegetation types. Please, clarify.
10, Fig.3. is clear and useful. Please call it as “flow chart”.
- Fig. 4. and Fig. 5 probably show all information properly. However, some rare habitat types and transitions were hard to see. Please, consider to present only a part of these maps with a magnified version just for a demonstration of changes. (Full maps can appear in appendix.) In fact, Table 6 and 7 present all related quantitative results.
- Figure 6 is excellent, nicely shows the major trends of vegetation changes.
- Result and Discussion. I suggest to separate these two sections. Please, avoid the simple repetition of numbers or trends that were already depicted in Figures and Tables. Focus on major changes and explain/interpret (!) the patterns detected. Please, cite similar studies about vegetation changes in riverine vegetation. Description of spatial aspects (vegetation zonation and other potential patterns) needs quantification and should be compared with other rivers. How the flood regime changed here over the study period? Please, discuss and evaluate. Separate spontaneous (natural) vegetation transitions from man-made changes.
14, “Land sanding degradation” has been mentioned. Please, describe this very important phenomenon with more details.
15, Closing paragraph is Discussion: It is not clear how the data in Table 6 and 7 related to human induced processes (urbanization, industrialization, agricultural modernization) in this region.
- Conclusions: Point 2 is not supported in the lack of quantitative analyses.
- Conclusions: Point 3 is not supported in the lack of data.
- Conclusions: Point 4 needs proof and reasoning.
In summary, the study based on valuable long-term data, used reliable methods and revealed nice patterns assessing changes in riverine vegetation over 30 years. However, the study context was poorly developed and causal mechanisms were poorly discussed. Results have not been compared to similar studies. Authors were not searching for general patterns. Practical aspects (implications for policy and land management) have not been elaborated.
The presentation of data and results need a very strong revision.